# Veterans Training Service Dogs for Other Veterans: An Animal-Assisted Intervention for Post-Traumatic Stress Disorder

**DOI:** 10.3390/bs15091180

**Published:** 2025-08-29

**Authors:** Cheryl A. Krause-Parello, Erika Friedmann, Deborah Taber, Haidong Zhu, Alejandra Quintero, Rick Yount

**Affiliations:** 1Division of Research, Florida Atlantic University, Boca Raton, FL 33431, USA; ckrausep@health.fau.edu; 2School of Nursing, University of Maryland Baltimore, Baltimore, MD 21201, USA; dtaber@umaryland.edu; 3Georgia Prevention Institute, Medical College of Georgia, Augusta University, Augusta, GA 30912, USA; hzhu@augusta.edu; 4Department of Biomedical Sciences, Florida Atlantic University, Boca Raton, FL 33431, USA; aquintero2014@fau.edu; 5Warrior Canine Connection, Boyds, MD 20841, USA; rick@warriorcanineconnection.org

**Keywords:** female veterans, telomere, cellular aging, heart rate variability, combat exposure

## Abstract

Research on the post-deployment reintegration needs of women veterans is limited. Non-traditional support may enhance mental health. Relationships with animals and volunteering may aid those with post-traumatic stress disorder (PTSD). Using the biopsychosocial model, we examined whether participation in an 8-week service dog training program (SDTP) affected telomere length (TL), heart rate variability (HRV), PTSD symptom severity, perceived stress, and anxiety in female veterans with PTSD, as well as whether combat exposure influenced these relationships. Female veterans (ages 32–72, M = 45.9, SD = 11.8) with PTSD were randomized to either the SDTP group (*n* = 13) or a comparison group (*n* = 15) that received dog training video content. The interventions lasted one hour weekly for 8 weeks. Outcomes were assessed pre-, mid-, and post-intervention. Linear mixed models with random intercepts examined changes from pre- to post-intervention and compared changes by group and combat exposure. TL changes differed [F(1,11.65) = 3.543, *p* = 0.085] by intervention. In the SDTP group, TL increased, indicating reduced cellular senescence (i.e., slower biological aging), whereas TL decreased in the CI group. Combat exposure moderated these changes [F(1,12.36) = 5.41, *p* = 0.038]. HRV changed by intervention group [F(1,389.08) = 10.623, *p* = 0.001]. HRV decreased (stress increased) in the SDTP group but not in the CI group. Combat exposure did not moderate HRV changes. PTSD symptom severity [F(1,48.04) = 19.22, *p* < 0.001], perceived stress [F(1,48.48) = 14.65, *p* < 0.001], and anxiety [F(1,47.30) = 6.624, *p* = 0.013] decreased significantly from pre- to post-interventions; the decreases did not differ by intervention or combat exposure.

## 1. Introduction

Female veterans represent a unique perspective on the effects of the military lifestyle; however, they have been underrepresented and understudied in research ([36]). Female veterans have always played a role in the military, in various forms ([32]). Historically, women served as nurses, clerks, and support staff, but over the decades their roles have expanded significantly to include leadership positions, combat assignments, and special operations. Although female veterans were granted permanent status in the military in 1948 with The Women’s Armed Services Integration Act, military research has remained predominantly focused on the experiences of male service members and veterans ([15]). However, rates of PTSD are significantly higher in female veterans compared to their male counterparts and the general population ([18]). Women often face higher rates of PTSD due to the nature of trauma exposure, particularly interpersonal traumas such as sexual assault and domestic violence ([18]). The literature indicates that differences in exposure to combat, pain, depression, PTSD, and military sexual trauma exist between the genders, according to data from Operation Enduring Freedom and Operation Iraqi Freedom ([16]; [33]). This has led female veterans to experience greater isolation than their male counterparts ([9]). There is subsequently an increased demand for research to inform clinical practice and build effective public policies for female veterans ([36]). Female veterans are often underrepresented in substance use disorder and intervention studies, emphasizing the necessity of providing access to evidence-based treatments tailored for this demographic due to the implications of trauma type on PTSD prevalence and treatment efficacy ([14]).

In this Special Issue, PTSD is recognized as a neurodivergent condition due to the profound and lasting changes that it can cause in brain structure and function. These changes can lead to symptoms that overlap with those seen in neurodivergent conditions (e.g., autism, ADHD), such as difficulties with emotional regulation and interpersonal relationships ([31]). Post-deployment reintegration continues to be difficult for both genders, as evidenced by the fact that the proportion of national daily deaths by suicide among veterans is greater than the proportion of the population of veterans ([7]). Nonetheless, utilizing different forms of social support may help to mitigate the mental health sequelae of service ([9]). The development and continuation of social support networks have been shown to reduce the severity of PTSD and be protective against suicide ([10]; [19]). There can be no doubt that the experience of serving in the military differs between most male and female service members; however, little research has been conducted on the unique post-deployment reintegration and readjustment demands that female veterans and their families face ([3]; [10]). Traditional interventions for PTSD involve a combination of general therapeutic approaches such as Cognitive Behavioral Therapy (CBT) and Prolonged Exposure (PE) therapy, medication, self-care strategies, and service dog programs ([23]; [26]; [35]). Indeed, most service dog programs for veterans involve the veteran training their own service dog. Other traditional interventions for PTSD specifically in female veterans include the SHE (Sexual Health Empowerment) program, which targets female veterans with a history of sexual trauma. However, the lives of female veterans are far from traditional ([9]), which may indicate that nontraditional or unexamined forms of social support may be especially beneficial. Similarly, there is increased effectiveness of alternative therapeutic approaches, including trauma-conscious yoga therapy, which demonstrated a notably higher treatment completion rate compared to traditional cognitive processing therapy ([37]). This suggests that traditional therapeutic modalities may not be fully meeting the needs of this unique veteran population.

Neurodivergent individuals often benefit from robust social support due to the challenges that they may face in social interactions and forming connections. Social support—whether from family, peers, community, or animals—has been linked to successful reintegration ([9]). Animals are known to provide humans with a sense of security and safety, social integration with individuals who have a common interest and concern for animals, the opportunity for nurturance, reassurance of worth from the human–animal relationship, and a sense of reliable alliance and relational consistency ([17]). A relationship developed with an animal differs from a relationship with a human due to the lack of judgment that exists between humans and animals (e.g., a person rarely feels that an animal is judging perceived inadequacies; [24]). These aspects may make relationships with animals particularly meaningful, especially in the case of returning veterans who may feel dissociated from loved ones to whom they return ([33]). Research on gender differences in pet attachment bonds reveals that gender is a significant predictor of the strength of pet–owner bonds, with women demonstrating a higher degree of attachment compared to men. This may be associated with women’s generally greater empathy levels, which frequently translate into deeper emotional connections with pets ([34]). Studies indicate that pet ownership encourages nurturing behaviors, with women’s perceptions of pets often aligning with their roles as caretakers and empathetic individuals ([2]). It has been posited that women cope with stress through caring for offspring, joining social networks, and organizing social gatherings in order to create an exchange of supportive resources and responsibilities ([6]). It may be that when a disruption in human social support occurs, individuals seek to replace that support through a relationship with an animal ([8]). This phenomenon may partially account for the number of young women choosing dog ownership over child-rearing ([1]). Furthermore, a significant percentage of pet owners, predominantly women, consider their pets to be integral to their happiness and social interactions, highlighting the relationship between the quality of pet attachments and mental health outcomes ([21]). Thus, animal attachment as a form of social support may be a manifestation of support-seeking among female veterans. However, not all female veterans may have the time, space, or means to care for their own dogs ([25]). This population often faces unique challenges, including higher rates of single parenthood, housing instability, or limited income during reintegration. The potential stressors associated with managing a service animal, including challenges in training and public integration, may create barriers to positive experiences ([26]). These factors can make pet ownership impractical or inaccessible.

Furthermore, a growing body of theoretical and empirical evidence reveals that volunteering and altruistic action—as long as they are not overwhelming to the individual—are associated with better psychological, mental, and physical health ([25]; [30]). Volunteerism may be beneficial in veterans’ reintegration into civilian life, providing a sense of purpose and replacing previous service-related roles while serving in active duty ([4]; [20]). Moreover, the veterans in this study volunteered to train a service dog for other veterans.

These relationships have not been fully understood in female veterans. Therefore, the objectives of this research were (1) to examine whether the SDTP led to changes in biological (telomere length, HRV) and psychological (PTSD symptom severity, perceived stress, anxiety) indicators of stress in female veterans with PTSD, and (2) to examine whether combat exposure moderates the above relationships.

## 2. Materials and Methods

### 2.1. Design

This randomized controlled trial with a repeated-measures design was guided by the biopsychosocial model ([11]).

### 2.2. Participants

The rights of human subjects were protected by receiving approval from a university institutional review board prior to data collection (University of Maryland Baltimore IRB HP-00083872). Military veterans were recruited from the community and the Warrior Canine Connection Program (WCC; http://warriorcanineconnection.org/, accessed on 1 February 2025; accredited by Assistance Dogs International) dog training waiting list. All the veterans met the WCC criteria for participation in their SDTP. The study’s inclusion criteria also included (1) female, (2) diagnosis of PTSD, and (3) able to give informed consent. The exclusion criteria for participation in the study were a fear of dogs, allergies to pet dander, active substance abuse, or history of animal abuse.

Once informed consent was signed, the participants were randomized to the SDTP or CI intervention group in blocks of 4. The sample consisted of 28 female military veterans with PTSD. Their ages ranged from 32 to 72 years (mean = 45.9, SD = 11.8). Thirteen participants were assigned to the SDTP group and fifteen to the CI group. The branches of the military represented included the Army (50%), Air Force (25%), Navy (10.7), and Marine Corps (7.1%). A plurality of the participants was White (60.7%), followed by Black or African American (28.6) and Other Race (10.7%). Most of the participants were married or in a domestic partnership (46.4%), followed by previously married (32.1%) and single (21.4%). Some of the participants had been engaged in direct combat (35.7%). More than half of the participants owned their own homes (60.7%); others rented (28.6%) or resided with a friend or family (10.7%). An overwhelming majority (78.6%) of the participants had a pet dog as a child, and 57.1% currently lived with a pet (e.g., dog or a cat). All the participants had at least a high school diploma or equivalent. There were no significant differences in the demographics of the two intervention groups (see Table 1).

### 2.3. Study Setting

This study took place at a service dog training facility in a semi-rural community near a major city in the mid-Atlantic region of the US. The participants were responsible for getting to and from the facility, which is not accessible by public transit, for all sessions. After a COVID-19-related pause in research, participants in the SDTP group conducted dog training sessions outdoors, rather than at the indoor training facility, and met the study team member for assessments at picnic tables or at their cars at the WCC site. Those in the CI group completed the video training in their homes and met the study team member outdoors at a community site or at their cars in parking lots for their assessments.

### 2.4. Study Interventions

#### 2.4.1. SDTP Intervention

Veterans randomly assigned to the dog training intervention worked closely with an experienced WCC Mission-Based Trauma Recovery (MBTR) Trainer (MBRT-T) to train a service dog for another veteran in need of a highly skilled service animal. Each of the eight training sessions lasted approximately one hour. The training followed the prescribed modules based on the Warrior Canine Connection Program (WCC; http://warriorcanineconnection.org/ accessed on 1 February 2025; accredited by Assistance Dogs International). The modules included the following: I. Warrior Ethos; II. The Human–Animal Bond; III. Shaping the Dog’s/Warrior’s Behavior; IV. Emotional and Physical Synchronization; V. Teaching New Commands; VI. Socialization; VII. Connecting to Family; and VIII. Gain thorough Loss.

#### 2.4.2. Control Intervention (CI)

Veterans randomly assigned to the CI participated in weekly online training sessions (https://e-trainingfordogs.com, accessed on 1 February 2025) delivered by experienced SD trainers. The module’s content was like the content provided in the WCC SDTP. Each training session was approximately one hour over the course of 8 weeks. The participants in the CI had no interaction with the dogs. However, at the end of the CI, participants were offered the opportunity to participate in the SDTP.

### 2.5. Study Assessments

Cellular aging was assessed by telomere length analysis obtained through saliva before the first and after the eighth intervention session for both groups. Saliva includes buccal cells and leukocytes suitable for use in telomere length analysis. An extra-gentle DNA extraction method was used to minimize oxidative damage to the DNA during the extraction process. We used the published protocol ([22]) and obtained absolute telomere length using ScienCell’s Absolute Human Telomere Length Quantification (AHTLQ) qPCR Assay Kit (ScienCell Research Laboratory, Carlsbad, CA, USA). AHTLQ is designed to directly measure the average telomere length of a human cell population. The telomere primer set recognizes and amplifies telomere sequences. The single-copy reference primer set recognizes and amplifies a 100 bp-long region on human chromosome 17 and serves as a reference for data normalization. The reference genomic DNA sample with known telomere length serves as a reference for calculating the telomere length of target samples. The comparative ΔΔCq (Quantification Cycle Value) method was used for quantification. The total telomere length of the target sample per diploid cell = reference sample telomere length × 2^−ΔΔCq^. There are 92 chromosome ends in one diploid cell; therefore, the average telomere length on each chromosome end was calculated as follows: total telomere length of the target sample/92. The lower limit of quantification was around 2 kilobase pairs (Kb), and the upper limit was around 20 kb. For quality control, the pre- and post-intervention DNA samples from the same participant were measured in the same plate in duplicates. If the standard deviation of Ct values was above 0.5, the samples from the same participants at both timepoints were repeated at the same time. Telomere length was positively skewed; therefore, natural log transformation was used to achieve normality. Higher values of the transformed value indicate less cellular aging. 

The heart rate variability (HRV) of veterans was measured with a Polar H10 heart rate sensor, worn around the chest, along with a matching Polar V800 GPS Sports Watch worn around a wrist to save data from the heart rate sensor. Polar monitors were applied prior to each intervention session and removed immediately afterward. HRV was not collected continuously throughout the 8-week study period; rather, recordings were obtained only during the in-person intervention sessions (i.e., the 8 h of training) at weeks 1, 4, and 8. Thus, the data reflect acute HRV recordings during training sessions, not the average HRV across the entire intervention period. Veterans were ambulatory during the intervention activities, and the duration of the sessions varied. To standardize HRV reporting, each intervention session was divided into eight equal segments. However, fewer segments were created when the average time per segment fell below three minutes. The HRV metric analyzed was the root-mean-square of successive differences (RMSSD). For each segment, average RMSSD values were downloaded from the Polar software. Polar employs proprietary algorithms for automatic filtering of RR intervals to remove outliers and improve data quality. While these algorithms are not publicly disclosed, studies suggest that Polar’s filtering methods are validated by comparison with ECG-generated data, except during intense exercise ([28]), which did not occur in the current study. HRV was positively skewed and, therefore, natural log-transformed to achieve normality. For the transformed outcome, higher values are better.

Post-traumatic stress disorder symptom severity (PTSDSS) was assessed prior to session 1 and after sessions 4 and 8 with the PTSD Checklist for DSM-5 (PCL-5). This 20-item scale is based on the DSM-5 criteria for PTSD. It demonstrates strong reliability (α = 0.94) and convergent validity (r = 0.74 to 0.85) ([5]; [13]). According to the U.S. Veterans Administration, scores of 31–33 or above indicate a need for increased PTSD care (Veterans Administration).

Perceived stress (PS) was assessed prior to session 1 and after sessions 4 and 8 with the Patient-Reported Outcomes Measurement Information System (PROMIS) Perceived Stress Fixed Form v2.0 (PROMIS, PS). This is a 10-item scale measuring one’s subjective experience of stress in the past month. Seventy-three items related to PS have demonstrated excellent reliability (Cronbach’s α= 0.91) and modest validity (coefficient range = 0.10–0.30) in the general population and meet the NIH Toolbox criteria ([27]).

Anxiety (A) was assessed prior to session 1 and after sessions 4 and 8 with the PROMIS Anxiety-Short Form 8a, an 8-item scale measuring self-reported fear and anxious misery. This scale has established reliability (α > 0.97) and validity ([29]).

### 2.6. Statistical Analysis

Data were cleaned and assumptions checked prior to analysis. Descriptive statistics were summarized for the entire sample and for the participants in the two groups. Chi-squared tests or ANOVAs were used to compare the characteristics of the groups. Missing data were completely at random (Little’s MCAR *p* = 0.826). Examination of the normality of outcomes revealed significant skews in telomere length and heart rate variability. These were corrected with natural log transformations.

Linear mixed models (LMMs) with random intercepts for participants were used to assess differences in changes in the biological and psychological stress outcome measures between the SDTP and CI group participants. Separate sets of analyses were conducted for each outcome. Fixed effects included group (SDTP or CI), week (1, 8), and their interaction. The hypothesized effect of the intervention was tested with the week-by-group interaction. Interactions of combat exposure with group and time were explored to provide information about its relationship to changes.

## 3. Results

We aimed to examine whether the SDTP led to changes in biological stress outcomes (telomere length and heart rate variability). Telomere length did not change significantly from before the first to after the last intervention session in the two groups combined. There was a tendency for a difference in changes between the two intervention groups [F(1,11.65) = 3.543, *p* = 0.085]. The trajectories showed that telomere length decreased (indicating increased aging) in the CI group and increased (indicating decreased aging) in the SDTP group from baseline to 8 weeks later. When combat exposure was examined as a moderator of changes in telomere length between the two groups, there was a significant moderation effect [F(1,12.36) = 5.41, *p* = 0.038; Figure 1]. Changes in telomere length from the beginning to the end of the intervention depended on intervention group and combat exposure. Telomere length decreased from week 1 to 8 in the CI group, with faster decreases among veterans who had combat exposure than those who did not have exposure. The largest improvement in telomere length was in the SDTP group members who had combat exposure, and the largest decrease in telomere length (increase in cellular aging) occurred in the CI group members who had combat exposure. Telomere length increased in the SDTP group, indicating that cellular aging decreased; in members of the SDTP group, the increases did not differ significantly according to combat exposure. Telomere length decreased in the CI group, indicating that cellular aging increased; in members of the CI group, the increases in cellular aging were greater in those who had experienced combat exposure than in those who had not.

Heart rate variability changed differently from before to after the intervention according to the intervention group [F(1,389.08) = 10.623, *p* = 0.001; Figure 2]. Heart rate variability decreased, indicating increased stress, in the SDTP intervention group, while it did not change in the CI group. Combat exposure did not moderate the relationship between changes in HRV from baseline to week 8 and intervention group.

We aimed to examine whether the SDTP led to changes in psychological (PTSD symptom severity, perceived stress, anxiety) indicators of stress in female veterans with PTSD, as well as whether combat exposure moderates these changes.

PTSD symptom severity decreased significantly from baseline to 8 weeks [F(1,48.04) = 19.22, *p* < 0.001]; however, the decrease did not differ significantly between the two intervention groups [F(1,48.04) = 0.327, *p* = 0.570].

Perceived stress decreased significantly from baseline to 8 weeks [F(1,48.48) = 14.65, *p* < 0.001]; however, the decrease did not differ significantly between the two intervention groups [F(1,48.48) = 0.001, *p* = 0.972]. Anxiety decreased significantly from baseline to 8 weeks [F(1,47.30) = 6.624, *p* = 0.013]; however, the decrease did not differ significantly between the two intervention groups [F(1,47.30) = 0.060, *p* = 0.808]. The decreases in PTSD symptom severity, perceived stress, and anxiety did not depend on the intervention group. Combat exposure did not moderate these changes.

## 4. Discussion

The current study is groundbreaking in two ways: (1) it focuses on the understudied group of female veterans with PTSD, and (2) it examines cellular aging in this population. The inclusion of both biological and psychological stress indicators in this population indicates that perceptions may not reflect physiological indicators of stress. Physiological stress indicators changed differently in the two intervention groups. The SDTP intervention reduced stress indicators, particularly in female veterans with combat exposure. The reduction in cellular aging, demonstrated by increased telomere length, provided initial evidence that non-pharmacological intervention can reduce cellular aging in female veterans. Both interventions led to decreases in psychological stress indicators; however, the decreases did not differ between the two intervention groups. This suggests the possibility that the attention that the female veterans received from being in the study may have provided support that lessened their perceptions of being stressed.

When examining the physiological outcome of HRV, we anticipated that the control group would show minimal change across sessions, whereas the intervention group might show evidence of adaptive shifts over time. Contrary to expectations, the intervention group exhibited decreases in RMSSD, while the control group remained stable, from week 1 to week 8. Our primary intent was not to compare absolute HRV values between groups but, rather, to explore patterns of within-group change over time. HRV is a highly sensitive physiological measure that can vary on a beat-to-beat basis and is susceptible to contextual influences. In our study, contextual stressors related to the introduction of revised COVID-19 safety protocols (e.g., mask wearing, physical distancing, outdoor data collection at picnic tables with limited privacy, changes in personnel contact when chest straps had to be self-placed) may have introduced unanticipated stress and variability into the HRV responses. These factors likely contributed to the unexpected directional change in HRV observed in the intervention group.

It is also possible that learning how to train a service dog led to increased interactions with their own dogs for the members of both intervention groups. Acquiring skills in positive reinforcement, reading pet body language, and establishing consistent routines may have enhanced the human–animal bond at home, contributing to secondary therapeutic benefits. Service dog training uniquely combines emotional healing with the formation of a close bond between veterans and their animals, offering therapeutic mechanisms distinct from those of broader volunteer activities ([26]). While volunteering activities typically allow for broader community engagement, service dog training intricately intertwines the veteran’s emotional healing with the process of forming a bond with the animal, providing distinct therapeutic mechanisms that other forms of volunteering may not offer.

Future studies could explore how dog training education affects personal pet relationships, whether these effects persist over time, and whether they contribute to long-term improvements in mental health, social engagement, or daily functioning among veterans with PTSD. The data point to the promise of human–animal relationships in supporting PTSD and related neurodivergent conditions through improved social and emotional engagement. Attachment to animals may promote empathy not only towards pets but also towards other humans ([12]). Additionally, this research underscores the value of integrating environmental, biological, and relational factors into holistic treatment strategies for neurodivergent populations.

The current study evaluated whether combat exposure was related to responses to the SDTP intervention among women. However, it did not evaluate the potential differential impact of other sources of trauma among the female veterans. Although we did not formally inquire about the sources of trauma for the study participants, several women volunteered that their PTSD stemmed from military sexual trauma. This opens a question of whether the source of the trauma is related to the effectiveness of the SDTP intervention. This would be an important area for future research, since one in every three female veterans reports sexual harassment or abuse (U.S. Department of Veterans Affairs).

### Limitations

The results of this study should be viewed with caution due to the small sample size and being underpowered. However, due to the small sample size, if the results support clinically meaningful significance or are trending in the right (positive) direction but not statistically significant, that does not mean that the outcomes are not worthy of consideration. We also acknowledge that eight weeks may be too short to detect the changes in telomere length, and the positive signal detected in our study needs to be validated in independent studies. We also agree that examining telomerase activity would be a better approach. However, no live cells were preserved in our study; therefore, we were unable to measure telomerase activity. Future studies examining telomerase activity are warranted. It should also be noted that because the intervention and control conditions necessarily involved different behavioral activities (ambulatory training exercises with dogs versus sitting to watch control videos), direct comparisons of absolute HRV values between groups are confounded. For this reason, the HRV findings should be interpreted as exploratory and focused on within-group change over time rather than between-group differences. Furthermore, this study was started before the COVID-19 pandemic and had to be halted in its entirety. Once the study resumed, the protocols had to be revisited and revised due to public safety health guidelines. The pandemic may have had a serious negative effect on stress levels that had nothing to do with the study itself. We nonetheless believe that these findings are important to report, as they provide valuable methodological insights for future researchers attempting to integrate HRV measurement into field-based interventions.

## 5. Conclusions

This study shed light on the effects of the SDTP on biological and psychological indicators of stress in female veterans with PTSD and combat exposure moderating these relationships. Female veterans who participated in the SDTP displayed a decrease in stress, and combat exposure moderated the changes. Therefore, it is suggested that future fully powered studies should examine the efficacy of service dog training programs as therapeutic interventions for female veterans with PTSD symptoms.

## Figures and Tables

**Figure 1 behavsci-15-01180-f001:**
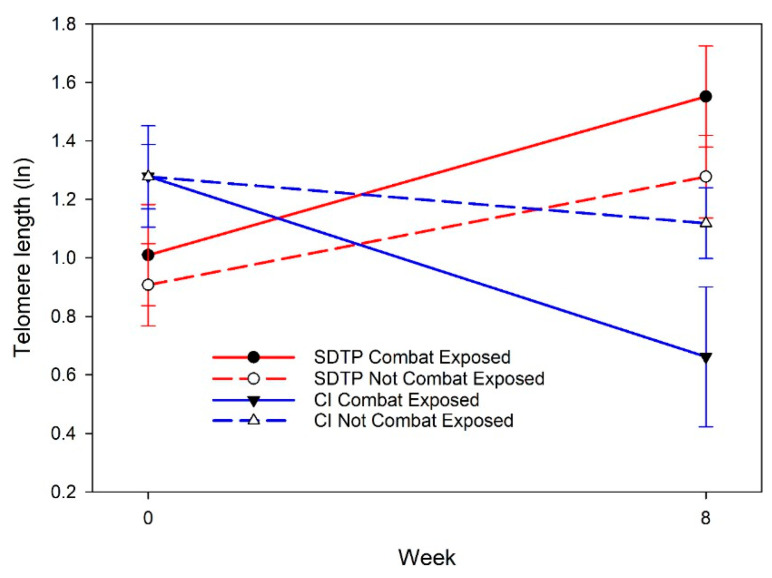
Changes in telomere length according to week, intervention group, and combat exposure estimated from analysis of linear mixed models. Lower telomere length indicates greater cellular aging. SDTP = service dog training program; CI = dog training video control intervention.

**Figure 2 behavsci-15-01180-f002:**
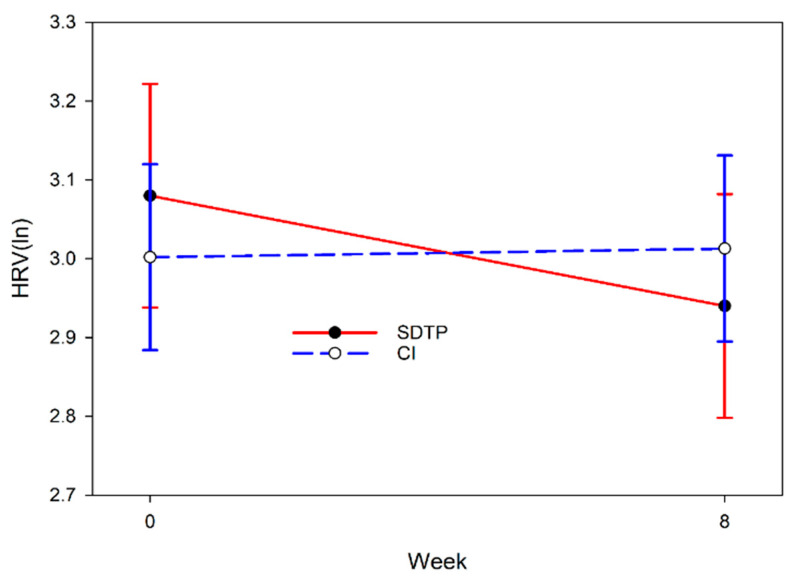
Changes in heart rate variability according to week and intervention group estimated from analysis of linear mixed models. Lower heart rate variability indicates less stress. SDTP = service dog training program; CI = dog training video control intervention.

**Table 1 behavsci-15-01180-t001:** Demographic descriptions of the participants according to random intervention group assignment: service dog training program (SDTP, N = 13), control (CI, N = 15).

All Freq		All %	SDTP Freq	SDTP %	CI Freq	CI %	(df) ChisSq	*p*=
Race							(2) 0.99	0.61
African American/Black	8	29.6	5	38.5	3	21.4		
White	17	63.0	7	53.8	10	71.4		
Other	2	7.4	1	7.7	1	7.1		
Ethnicity							(1) 0.23	0.63
Non-Hispanic	25	89.3	12	92.3	13	86.7		
Hispanic	3	10.7	1	7.7	2	13.3		
Education							(4) 4.29	0.37
Some College	3	10.7	2	15.4	1	6.7		
Associates	3	10.7	1	7.7	2	13.3		
Bachelors	6	21.4	4	30.8	2	13.3		
Masters	13	46.4	6	46.2	7	46.7		
Doctorate	3	10.7	0	0	3	20.0		
Marital Status							(2) 0.61	0.45
Married/Partner	13	46.4	6	46.2	7	46.7		
Never Married	6	21.4	4	30.8	2	13.3		
Previously Married	9	32.1	3	23.1	6	40.0		
Residence							(2) 0.25	0.88
Rent	8	28.6	4	30.8	4	26.7		
Own	17	60.7	8	61.5	9	60.0		
Family/Friend	3	10.7	1	7.7	2	13.3		
Has Children							(1) 2.23	0.14
No	18	64.3	8	61.5	10	66.6		
Yes	10	35.7	5	38.5	5	33.3		
Lives with a Pet							(1) 0.28	0.60
No Pet	3	15.8	1	11.1	2	20.0		
Pet	16	67.9	9	88.9	8	80.0		
Dog when Child							(1) 0.05	0.82
No Dog	5	17.9	2	16.7	3	20.0		
Dog	22	78.6	10	83.3	12	80.0		
Service Branch							(3) 3.70	0.34
Air Force	7	25.0	2	18.2	5	33.3		
Army	14	50.0	6	54.4	8	53.3		
Navy	3	10.7	1	9.1	2	13.3		
Marines	2	7.1	2	18.2	0	0.0		
Initial Military Classification							(1) 2.07	0.15
Enlisted	20	71.4	11	84.6	9	60.0		
Officer	8	28.6	2	15.4	6	40.0		
Direct Combat Exposure							(1) 0.61	0.44
No	18	64.3	9	69.2	9	60.0		
Yes	10	35.7	4	30.8	6	40.0		
Age (Years)	45.9	11.8	43.2	8.1	48.3	14.0	(1, 26) 1.37	0.253
PTSDSS * at entry	43.6	16.7	47.2	17.2	41.3	16.6	(1, 26) 0.60	0.444

* PTSDSS = Score on the PCL-5.

## Data Availability

The raw data supporting the conclusions of this article will be made available by the authors upon request.

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
