# Peer review of "Veterans Training Service Dogs for Other Veterans: An Animal-Assisted Intervention for Post-Traumatic Stress Disorder"

_behavsci, 2025, doi:10.3390/bs15091180_

Round 1

Reviewer 1 Report

Comments and Suggestions for Authors

Thank you for the opportunity to review this interesting and ambitious study comparing an 8 week service dog training program to a control dog training video comparison in women veterans with PTSD.  They used a number of outcome measures, including objective biological variables (telomere length, HRV) as well as subjective measures (perceived stress).  While I am impressed by the overall design, the paper would require major revisions to be considered for publication.

Overall, I think the introduction needs to be tightened significantly.  The authors are trying to be too broad (e.g. we don't need to know the WHO definition of mental health).  Please focus the introduction quickly on the experience of women veterans with PTSD and the need to test interventions with that group.  Some of the literature cited is grossly outdated (e.g. Weiss 1974, Pollitt 1992), sexist in its assumptions, relying on outdated stereotypes and wildly inappropriate for an article on women veterans.  Please cite appropriate empirical literature about women veterans and PTSD such as:

Mouilso, E. R., Tuerk, P. W., Schnurr, P. P., & Rauch, S. A. (2016). Addressing the gender gap: Prolonged exposure for PTSD in veterans. Psychological Services13(3), 308.

Stefanovics, E. A., & Rosenheck, R. A. (2020). Gender differences in outcomes following specialized intensive PTSD treatment in the Veterans Health Administration. Psychological Trauma: Theory, Research, Practice, and Policy12(3), 272.

I also find the use of the term "neurodivergent" to describe individuals with PTSD to be very odd and inappropriate.  Neurodivergence refers to developmental disorders (Autism, ADHD) NOT to acquired disorders that can impact brain structure, brain function or psychosocial functioning (PTSD, TBI, Dementia).  Please cut that term from the text and use appropriate nosological terminology.   

Overall, the introduction needs to position the paper much more appropriately by citing existing literature on interventions for PTSD generally, interventions for women veterans, and prior use of service dog training interventions  in this population.  See for example:

Whitworth, J. D., Scotland-Coogan, D., & Wharton, T. (2019). Service dog training programs for veterans with PTSD: Results of a pilot controlled study. Social Work in Health Care58(4), 412-430.

With respect to the methods, I would prefer to have more detail about the method used for establishing telomere length (i.e. qPCR, Southern blot, Flow-FISH).  8 weeks is generally far too short a time to detect changes in telomere length.  Examining telomerase activity would have been a better approach.  I also want more detail about HRV.  How was heart rate acquired?  What was the length of time?  Supine v standing versus ambulatory?  What corrections were used to detect and eliminate artifacts?  Exactly what measure of HRV is reported?  SDNN?  H/L Frequency?  It's impossible to evaluate in the current manuscript, and all of these details matter.

It's also unclear to me how much impact 8 hours over 8 weeks of working with a dog (the same dog?  a different dog every time?) could realistically have.  Typical service dog training programs are much more intensive, with the training handlers spending most of the day with the same dog over multiple weeks.  Indeed, most service dog programs for veterans involve the veteran training their OWN service dog. 

There is significant literature on this that the authors don't cite.

e.g. LaFollette, M. R., Rodriguez, K. E., Ogata, N., & O'Haire, M. E. (2019). Military veterans and their PTSD service dogs: associations between training methods, PTSD severity, dog behavior, and the human-animal bond. Frontiers in veterinary science6, 431718.

Leighton, S. C., Nieforth, L. O., & O’Haire, M. E. (2022). Assistance dogs for military veterans with PTSD: A systematic review, meta-analysis, and meta-synthesis. PLoS One17(9), e0274960.

Rodriguez, K. E., LaFollette, M. R., Hediger, K., Ogata, N., & O’Haire, M. E. (2020). Defining the PTSD service dog intervention: Perceived importance, usage, and symptom specificity of psychiatric service dogs for military veterans. Frontiers in psychology11, 519718.

O'haire, M. E., & Rodriguez, K. E. (2018). Preliminary efficacy of service dogs as a complementary treatment for posttraumatic stress disorder in military members and veterans. Journal of consulting and clinical psychology86(2), 179.

Kloep, M. L., Hunter, R. H., & Kertz, S. J. (2017). Examining the effects of a novel training program and use of psychiatric service dogs for military-related PTSD and associated symptoms. American Journal of Orthopsychiatry87(4), 425.

Reviewer 2 Report

Comments and Suggestions for Authors

I very much enjoyed reviewing this manuscript, and found the subject to be really interesting. I hope that these comments are found to be constructive and useful, rather than critical.

Abstract:

Page 1, Line 22: rephrase information regarding the comparison intervention for clarity, as it is a bit confusing as it reads right now.

Page 1, Line 26-27: rephrase this sentence “TL increased (decreased senescence) in the SDTP and decreased in the CI” for clarity and define “decreased senescence”.

Introduction:

Page 1, Lines 40-41: a bit more could be said about this sentence “Women veterans have always had a role in the military, just in various forms (Smith, 2022).” Would you be able to share a bit more about what various forms women veterans have had roles?

Page 3, Line 94: this line should read “feels that an animal is judging perceived inadequacies; Pickering, 2003)” for appropriate reference formatting.

Page 3, first paragraph generally: this paragraph could be strengthened by a discussion of gender differences related to pet attachment bonds. Existing literature has found that women tend to be more strongly bonded to their companion animals than men, and this might be useful to incorporate here.

Page 3, Lines 108-109: I think that this statement could be elaborated on a bit more “However, not all women veterans may have the time, space or means to care for their own dogs (Post, 2011).” What unique challenges does this population face that would make caring for a dog more difficult?

Materials and Methods:

Page 3, Line 130: remind the reader of what WCC is an abbreviation for

Page 5, Line 195: insert a comma after “sensor” for better sentence clarity

Results:

Page 5, Line 235: insert a semi-colon after “CI group” for better sentence clarity

Discussion:

Rephrase the first paragraph for clarity to the following: “The current study is groundbreaking in two ways: 1) it focuses on the understudied group of women veterans with PTSD; and 2) it examines cellular aging in this population.

More is needed in the discussion of how these results relate to existing literature. It may be particularly interesting to discuss this in terms of human-animal attachment, or in terms of other volunteer experiences of veterans (why is this type of volunteering different and how is it similar?).

Page 8, Lines 293-295: This is interesting: “It is also possible that learning how to train a service dog led to increased interactions with their own dogs for the members of both intervention groups.” I wonder if more could be said here about this, or whether a future direction could be discussed in terms of this statement.

Round 2

Reviewer 1 Report

Comments and Suggestions for Authors

The paper is much improved, but still didn't address how Telomere length was calculated, or the metric for HRV that was used. Also it's still completely unclear when exactly HRV was captured. "during intervention" could mean for the full 8 weeks. or only during the 8 hours of the actual intervention. HRV would be completely confounded by ambulatory exercise during training v sitting watching videos during the control intervention. As a result, any changes differences in HRV seem uninterpretable to me. That whole section needs to be explained much better and the limitations need to be made clear (especially since the results were the opposite of what was predicted, and inconsistent with all the other findings. 
